# Greedy-based Value Representation for Efficient Coordination in Multi-agent Reinforcement Learning

## Abstract

Due to the representation limitation of the joint Q value function, multi-agent reinforcement learning (MARL) methods with linear or monotonic value decomposition suffer from the relative overgeneralization. As a result, they can not ensure the optimal coordination, leading to instability and poor performance. Existing methods address the relative overgeneralization by achieving complete expressiveness or learning a bias, which are insufficient to solve the problem. In this paper, we propose the optimal consistency, a criterion to evaluate the optimality of coordination. To achieve the optimal consistency, we introduce the True-Global-Max (TGM) condition for linear and monotonic value decomposition, where the TGM condition can be ensured when the optimal stable point is the unique stable point. Therefore, we propose the greedy-based value representation (GVR) to ensure the optimal stable point via inferior target shaping and eliminate the non-optimal stable points via superior experience replay. We conduct experiments on various benchmarks, where GVR significantly outperforms state-of-the-art baselines. Experiment results demonstrate that our method can ensure the optimal consistency under sufficient exploration.

## 1 Introduction

By taking advantage of the deep learning technique, cooperative multi-agent reinforcement learning (MARL) shows great scalability and excellent performance on challenging tasks (Vorotnikov et al., 2018; Wu et al., 2020) such as StarCraft unit micromanagement (Foerster et al., 2018). As an efficient paradigm of cooperative MARL, centralized training with decentralized execution (CTDE) (Oliehoek et al., 2008; Foerster et al., 2016; Lowe et al., 2017) gains growing attention. A simple and effective approach to adopt CTDE in value-based cooperative MARL is linear value decomposition (LVD) or monotonic value decomposition (MVD). However, both LVD and MVD suffer from relative overgeneralization (Panait et al., 2006; Wei et al., 2018) due to the representation limitation of the joint Q value function. As a result, they can not guarantee optimal coordination.

Recent works address the problem from two different perspectives. The first kind of method aims to solve the representation limitation directly through value functions with complete expressiveness capacity (e.g., QTRAN (Son et al., 2019) and QPLEX (Wang et al., 2020)). However, learning the complete expressiveness is impractical in complicated MARL tasks because the joint action space increases exponentially with the number of agents. The other kind of method tries to overcome relative overgeneralization by learning a bias (e.g., WQMIX (Rashid et al., 2020) and MAVEN (Mahajan et al., 2019)), which lacks theoretical and quantitative analysis of the problem and is only applicable in specific tasks. As a result, these methods are insufficient to guarantee optimal coordination. More discussions about related works are provided in Appendix A.

Value decomposition is a popular approach to assign credit for individual agents in fully cooperative MARL tasks, where the main concern is the optimality of coordination. In a **successful case** of credit assignment via value decomposition, individual agents act according to their local policies and achieve the best team's performance. To evaluate the value decomposition, we propose the **optimal consistency**, a criterion concerning the optimality of coordination. The optimal consistency can be decomposed into two conditions: Individual-Global-Max (IGM) and **True-Global-Max (TGM)**.

In this paper, to achieve the optimal consistency efficiently, we investigate the requirements of the TGM condition and go deep into the mechanism of the value representation for LVD and MVD, where the IGM condition always holds. We first derive the expression of the joint Q value function of LVD and MVD, by which we draw some interesting conclusions. **Firstly**, LVD and MVD share the same expression of the joint Q value function. **Secondly**, the correspondence between the joint greedy action and the maximal Q true value (i.e., the team's best performance) depends heavily on the **task-specific** reward function for LVD and MVD. **Thirdly**, there may be multiple stable points of the joint greedy action. As a result, the joint policy may converge to different results. More importantly, in some of the stable points, the true Q value of the joint greedy action is not maximal, which is the **root cause** of non-optimal coordination and relative overgeneralization. To ensure the TGM condition for LVD and MVD, the stable point satisfying the TGM condition (we call it the optimal stable point) is required to be the **unique** stable point, which is the target problem to be solved in this paper.

To solve the target problem, we propose the greedy-based value representation (GVR). According to previous conclusions, the stable points are task-specific due to their dependency on the reward function, for which we propose the inferior target shaping (ITS). ITS dynamically modifies the true Q value of inferior samples (i.e., the samples worse than the current greedy) according to current greedy Q value, which is theoretically proved to stabilize the optimal point under any reward function. Besides, under ITS, the stability of a non-optimal point depends only on the probability ratio of superior samples (i.e., the samples better than the current greedy) to the non-optimal sample, where the non-optimal stable points can be eliminated under a large enough ratio (Eq.7). We prove two simple ways applied by previous works (i.e. applying weight on the superior samples (Rashid et al., 2020) and improving exploration (Mahajan et al., 2019)) are both inapplicable to raise the ratio because the probability of superior samples decreases exponentially with the number of agents. Therefore, we further propose the superior experience replay (SER), which achieves **almost constant probability** of superior samples by saving them in a superior buffer. SER is theoretically proved to eliminate the non-optimal stable points under ITS.

We have three contributions in this work. (1) This is the first work to derive the exact expression of the joint Q value function for LVD and MVD. (2) We point out the root cause of non-optimal coordination and further propose the target problem to be solved for LVD and MVD. (3) We propose the GVR method, which is proved theoretically to ensure the optimal consistency under sufficient exploration, and our method outperforms state-of-the-art baselines in various benchmarks.

## 2 PRELIMINARIES

### 2.1 DEC-POMDP

We model a fully cooperative multi-agent task as a decentralized partially observable Markov decision process (Dec-POMDP) described by a tuple $\mathcal{G} = < S, U, P, r, Z, O, n, \gamma >$ (Guestrin et al., 2001; Oliehoek & Amato, 2016). $s \in S$ denotes the true state of the environment. At each time step, each agent $a \in A \equiv \{1, 2, \cdots, n\}$ receives a local observation $z^a \in Z$ produced by the observation function $O : S \times A \to Z$, and then chooses an individual action $u^a \in U$ according to a local policy $\pi^a(u^a|\tau^a) : T \times U \to [0, 1]$, where $\tau^a \in T \equiv (Z \times U)^*$ denotes the local action-observation history. The joint action of $n$ agents $\mathbf{u}$ results in a shared reward $r(s, \mathbf{u})$ and a transition to the next state $s' \sim P(\cdot|s, \mathbf{u})$. $\gamma \in [0, 1)$ is a discount factor.

We denote the joint variable of group agents with bold symbols, e.g., the joint action $\boldsymbol{u} \in \boldsymbol{U} \equiv U^n$, the joint action-observation history $\boldsymbol{\tau} \in \boldsymbol{\mathcal{T}} \equiv T^n$, and the **joint interactive policy** (i.e., the policy interacts with environment to generate trajectories) $\boldsymbol{\pi}(\mathbf{u}|\boldsymbol{\tau})$. The **true Q value** of $\boldsymbol{\pi}(\mathbf{u}_t|\boldsymbol{\tau}_t)$ is denoted by $\mathcal{Q}^{\boldsymbol{\pi}}(s_t, \boldsymbol{u}_t) = \mathbb{E}_{s_{t+1:\infty}, \boldsymbol{u}_{t+1:\infty}}[R_t|s_t, \boldsymbol{u}_t]$, where $R_t = \sum_{i=0}^{\infty} \gamma^i r_{t+1}$ is the discounted return. The action-state value function of agent $a$ and the group of agents are defined as **utility function** $\mathcal{U}^a(u^a, \tau^a)$ and **joint Q value function** $Q(\boldsymbol{u}, \boldsymbol{\tau})$ respectively. The true Q value is the target of the joint Q value in training, serving as the unique external criterion of the team's performance. The **greedy action** $\boldsymbol{u}^* := argmax_{\boldsymbol{u}} Q(\boldsymbol{u}, \boldsymbol{\tau})$ is defined as the joint action with the maximal joint Q value . The **optimal action** $\boldsymbol{u}_{opt} := argmax_{\boldsymbol{u}} \mathcal{Q}(s, \boldsymbol{u})$ is defined as the joint action with the best team's performance. For brevity, we sometimes omit the prefix "joint" for the joint variables.

### 2.2 Optimal Consistency and TGM Condition

In CTDE paradigm, agents are expected to act individually according to their local policies (i.e., the individual greedy actions) while achieve the optimal coordination (i.e., the maximal true Q value). Here we define the correspondence between the individual greedy actions and the maximal true Q value as the optimal consistency.

**Definition 1 (Optimal consistency).** Given a set of utility functions $\{\mathcal{U}^1(u^1, \tau^1)), \cdots, \mathcal{U}^n(u^n, \tau^n)\}$, and the true Q value $\mathcal{Q}(s, \boldsymbol{u})$, if the following holds

$$\{\underset{u^1}{argmax}\, \mathcal{U}^1(u^1, \tau^1), \cdots, \underset{u^n}{argmax}\, \mathcal{U}^n(u^n, \tau^n)\} = \underset{\boldsymbol{u}}{argmax}\, \mathcal{Q}(s, \boldsymbol{u}) \tag{1}$$

then we say the set of utility functions $\{\mathcal{U}^1(u^1, \tau^1)), \cdots, \mathcal{U}^n(u^n, \tau^n)\}$ satisfies the optimal consistency. For simplicity, we ignore situations with non-unique optimal actions.

The optimal consistency can be decomposed into two conditions: Individual-Global-Max (IGM) and True-Global-Max (TGM). The IGM condition proposed by QTRAN (Son et al., 2019) is defined on the correspondence between individual greedy actions and the joint greedy actions (formally, $\{argmax_{u^1}\, \mathcal{U}^1(u^1, \tau^1), \cdots, argmax_{u^n}\, \mathcal{U}^n(o^n, \tau^n)\} = argmax_{\boldsymbol{u}} Q(\boldsymbol{u}, \boldsymbol{\tau})$). To achieve the optimal consistency, the correspondence between the joint greedy action and the maximal true Q value is required, for which we define the TGM condition:

**Definition 2 (TGM).** Given a joint value function $Q(\boldsymbol{u}, \boldsymbol{\tau})$, and the true Q value $\mathcal{Q}(s, \boldsymbol{u})$, if the following holds

$$\underset{\boldsymbol{u}}{argmax}\, Q(\boldsymbol{u}, \boldsymbol{\tau}) = \underset{\boldsymbol{u}}{argmax}\, \mathcal{Q}(s, \boldsymbol{u}) \tag{2}$$

then we say the joint value function $Q(\boldsymbol{u}, \boldsymbol{\tau})$ satisfies the TGM condition. For simplicity, we ignore situations with non-unique optimal actions.

## 3 Investigation of the TGM Condition for LVD & MVD

Linear value decomposition (LVD) and monotonic value decomposition (MVD) are simple and naturally meet the IGM condition. To achieve the optimal consistency, we investigate the requirements of the TGM condition for LVD and MVD. According to Def.2, the TGM condition is related to the joint Q value function $Q(\boldsymbol{u}, \boldsymbol{\tau})$. In this section, we first derive the expression of $Q(\boldsymbol{u}, \boldsymbol{\tau})$ for LVD and MVD under $\epsilon-$greedy visitation. The expression indicates there may be non-optimal stable points that violate the TGM condition, which is the root cause of non-optimal coordination.

### 3.1 EXPRESSION OF THE JOINT Q VALUE FUNCTION FOR LVD & MVD

Firstly, take two-agent linear value decomposition as an example, where the joint Q value function $Q(u_i^1, u_j^2, \boldsymbol{\tau})$ is linearly factorized into two utility functions $Q(u_i^1, u_j^2, \boldsymbol{\tau}) = \mathcal{U}^1(u_i^1, \tau^1) + \mathcal{U}^2(u_j^2, \tau^2)$. $u_i^1, u_j^2 \in \{u_1, \cdots, u_m\}$ denote the individual actions of agent 1,2 respectively, where $\{u_1, \cdots, u_m\}$ is the discrete individual action space. Specially, we denote the individual greedy action of agent 1,2 with $u_{i*}^1, u_{j*}^2$ respectively. For brevity, $Q(u_i^1, u_j^2, \boldsymbol{\tau})$ and $\mathcal{U}^a(u_i^a, \tau^a)$ are represented by $Q_{ij}$ and $\mathcal{U}_i^a (a \in \{1,2\})$ respectively. Through the derivation provided in Appendix B.1, $Q_{ij}$ can be represented by the true Q values as

$$Q_{ij} = \frac{\epsilon}{m} \sum_{k=1}^m (\mathcal{Q}_{ik} + \mathcal{Q}_{kj}) + (1-\epsilon)(\mathcal{Q}_{i*j} + \mathcal{Q}_{ij*}) - \frac{\epsilon^2}{m^2} \sum_{i=1}^m \sum_{j=1}^m \mathcal{Q}_{ij}$$
$$- \frac{\epsilon(1-\epsilon)}{m} \sum_{k=1}^m (\mathcal{Q}_{i*k} + \mathcal{Q}_{kj*}) - (1-\epsilon)^2 \mathcal{Q}_{i*j*}$$

(3)

Verification of the expression is provided in Appendix B.2. For monotonic value decomposition, the expression is identical to Eq.3 (the proof is provided in Appendix C), which indicates the coefficients and bias on utility functions do not affect the joint Q value function. For situations with more than two agents, by referring to the derivation in Appendix B.1 and C, the expression of joint Q values can also be obtained.

### 3.2 REQUIREMENTS TO ENSURE THE TGM CONDITION FOR LVD & MVD

According to Eq.3, **the joint Q values change with the greedy action** $\{u_{i*}^1, u_{j*}^2\}$. An example of the joint Q values under 3 different $\{u_{i*}^1, u_{j*}^2\}$ is shown in Tab.1(b,c,d), where $\epsilon = 0.2$.

| 8(optimal) | -12 | -12 |
|---|---|---|
| -12 | 0 | 0 |
| -12 | 0 | 6 |

(a) The given $\mathcal{Q}_{ij}$.

| **7.40** | -8.33 | -7.93 |
|---|---|---|
| -8.33 | -24.06 | -23.66 |
| -7.93 | -23.66 | -23.26 |

(b) $Q_{ij}$ under $\{u_{i*}^1, u_{j*}^2\} = \{0, 0\}$.

| -19.90 | -10.04 | -9.64 |
|---|---|---|
| -10.04 | -0.17 | 0.23 |
| -9.64 | 0.23 | **0.63** |

(c) $Q_{ij}$ under $\{u_{i*}^1, u_{j*}^2\} = \{1, 1\}$.

| -24.38 | -14.52 | -9.32 |
|---|---|---|
| -14.52 | -4.65 | 0.55 |
| -9.32 | 0.5 | **5.75** |

(d) $Q_{ij}$ under $\{u_{i*}^1, u_{j*}^2\} = \{2, 2\}$.

Table 1: (a) The given true Q values $\mathcal{Q}_{ij}(u_i^1, u_j^2 \in \{0, 1, 2\})$. (b,c,d) Calculation results of the joint Q values $Q_{ij}$ with LVD or MVD under different greedy actions $\{u_{i*}^1, u_{j*}^2\}$, where $\{u_{i*}^1, u_{j*}^2\}$ is marked with a pink background. We denote the maximal joint Q values with numbers in bold. Due to the representation limitation, $Q_{ij} \neq \mathcal{Q}_{ij}$. Instead, $Q_{ij}$ overgeneralizes to $\mathcal{Q}_{ij}$.

Notice that the example in Tab.1(c) is an **unstable point**, because $\{u_{i*}^1, u_{j*}^2\} = \{1, 1\} \neq argmax\, Q_{ij} = \{2, 2\}$. To explain this, assume $\{u_{i*}^1, u_{j*}^2\}$ and $Q_{ij}$ of iteration $t$ are $\{u_{i*}^1, u_{j*}^2\}_t$ and $Q_{ij,t}$ respectively, where $\{u_{i*}^1, u_{j*}^2\}_t$ ($\{u_{i*}^1, u_{j*}^2\}$ in Tab.1(c)) $= argmax\, Q_{ij,t} = \{1, 1\}$. After training, $\{u_{i*}^1, u_{j*}^2\}_{t+1} = argmax\, Q_{ij,t+1}$ ($Q_{ij}$ in Tab.1(c)) $= \{2, 2\}$, which means $\{u_{i*}^1, u_{j*}^2\}_t$ **changes with iteration**. Since $Q_{ij,t+1}$ is related to $\{u_{i*}^1, u_{j*}^2\}_t$, both $\{u_{i*}^1, u_{j*}^2\}_t$ and $Q_{ij,t+1}$ are unstable. As a result, Tab.1(c) finally converges to the stable point in Tab.1(d). Here we define the stable point of LVD and MVD.

**Definition 3 (Stable point of LVD and MVD).** Given the joint Q value function $Q(\boldsymbol{u}, \boldsymbol{\tau})$ which has converged under LVD (or MVD) with the target $\mathcal{Q}(s, \boldsymbol{u})$, where $\mathcal{Q}(s, \boldsymbol{u})$ is the true Q value, and given an

interactive policy $\boldsymbol{\pi}(\boldsymbol{u}, \boldsymbol{\tau})$ with the greedy action $\boldsymbol{u}^*$ ($\boldsymbol{u}^* = argmax\ \boldsymbol{\pi}(s, \boldsymbol{u})$), if the following holds

$$\underset{\boldsymbol{u}}{argmax}\ Q(\boldsymbol{u}, \boldsymbol{\tau}) = \boldsymbol{u}^* \tag{4}$$

then we say $Q(\boldsymbol{u}, \boldsymbol{\tau})$ and $\boldsymbol{u}^*$ are stable, and this is a stable point of LVD (or MVD).

According to Def.3, the examples in Tab.1(b) ($argmax\ Q_{ij} = \{u_{i*}^1, u_{j*}^2\} = \{0, 0\}$) and Tab.1(d) ($argmax\ Q_{ij} = \{u_{i*}^1, u_{j*}^2\} = \{2, 2\}$) are both stable points (i.e., the joint Q values and the greedy action may converge to Tab.1(b) or Tab.1(d)). However, the stable points in Tab.1(d) **violates the TGM condition** (i.e., $argmax\ Q_{ij} \neq argmax\ \mathcal{Q}_{ij}$), which is the root cause of non-optimal coordination. Tab.1(b) and Tab.1(d) are defined as **non-optimal stable point** and **optimal stable point** respectively. To ensure the TGM condition, the optimal stable point is required to be the unique stable point. More discussion about the stable points is provided in appendix H.

## 4 METHOD

In this section, we introduce the greedy-based value representation (GVR). We first propose inferior target shaping (ITS) to ensure the optimal stable point under any reward function. Besides, under ITS, the stability of a non-optimal point depends only on the probability ratio of superior samples to the non-optimal sample. To eliminate the non-optimal stable points, we then investigate two simple methods (i.e., improving exploration and applying a weight on the superior samples) which raise the ratio. However, both methods are proved inapplicable due to the extremely small probability of superior samples under large joint action space. Finally, we describe superior experience replay, which achieves an almost constant proportion of superior samples through a superior buffer.

### 4.1 INFERIOR TARGET SHAPING

According to Eq.3, the joint Q value of any action is related to the true Q values of the whole joint action space for LVD and MVD, which indicates the stable points depend heavily on the reward function (proofs are provided in Appendix H). Firstly, we consider to remove the dependency. Because the exact true Q values of non-optimal samples are uninformative, we only represent the "non-optimality" of these samples. Here we propose the ITS target $\mathcal{Q}_{its}(s, \boldsymbol{u})$

$$\mathcal{Q}_{its}(s, \boldsymbol{u}) = \begin{cases} Q(\boldsymbol{u}^*, \boldsymbol{\tau}) - \alpha|Q(\boldsymbol{u}^*, \boldsymbol{\tau})| & \mathcal{Q}(s, \boldsymbol{u}) < Q(\boldsymbol{u}^*, \boldsymbol{\tau})\ and\ \boldsymbol{u} \neq \boldsymbol{u}^* \\ \mathcal{Q}(s, \boldsymbol{u}) & others \end{cases} \tag{5}$$

where $\boldsymbol{u}^*$ is the joint greedy action and $\alpha \in (0, 1]$. A large enough $\alpha$ prevents the confusion between greedy and inferior samples. The samples in the first case are called **inferior samples**. Besides, the action samples $\boldsymbol{u}$ which satisfy $\mathcal{Q}(s, \boldsymbol{u}) > \mathcal{Q}(s, \boldsymbol{u}^*)$ are called **superior samples**. ITS target only remains the information of non-optimality for inferior samples, which eases representation. Under ITS, given the greedy action $\boldsymbol{u}^*$ and any action $\boldsymbol{u}_s(\boldsymbol{u}_s \neq \boldsymbol{u}^*)$, assuming $\mathcal{Q}(s, \boldsymbol{u}^*) > 0$, we have

$$\Delta Q(\boldsymbol{u}_s, \boldsymbol{\tau}) = Q(\boldsymbol{u}_s, \boldsymbol{\tau}) - Q(\boldsymbol{u}^*, \boldsymbol{\tau}) = n(\eta_1 - \eta_2)\left[\mathcal{Q}(s, \boldsymbol{u}^*) - (1 - \alpha)Q(\boldsymbol{u}^*, \boldsymbol{\tau})\right] + n\eta_1 e_Q \mathcal{Q}(s, \boldsymbol{u}^*) \tag{6}$$

where $e_Q = \frac{\mathcal{Q}(s, \boldsymbol{u}_s) - \mathcal{Q}(s, \boldsymbol{u}^*)}{\mathcal{Q}(s, \boldsymbol{u}^*)}, \eta_1 = (\frac{\epsilon}{m})^{n-1}$ and $\eta_2 = (1 - \epsilon + \frac{\epsilon}{m})^{n-1}$. We provide two different versions of proof for Eq.6 in Appendix D, and the calculation result is verified in the experimental part (Fig.1). It is proved in Appendix D.1 that **there is always an optimal stable point under ITS**.

When $\Delta Q(\boldsymbol{u}_s, \boldsymbol{\tau}) > 0$, $argmax_{\boldsymbol{u}}Q(\boldsymbol{u}, \boldsymbol{\tau}) \neq \boldsymbol{u}^*$, which suggests current greedy action is unstable. If $\boldsymbol{u}^*$ is a non-optimal action, to destabilize it, let $\Delta Q(\boldsymbol{u}_s, \boldsymbol{\tau}) > 0$ and assume $Q(\boldsymbol{u}^*, \boldsymbol{\tau}) \approx \mathcal{Q}(s, \boldsymbol{u}^*)$ (this assumption is quite accurate, as verified in Appendix E.2), we have

$$\frac{\eta_1}{\eta_2} > \frac{\alpha}{\alpha + e_Q} \tag{7}$$

which indicates the non-optimal stable points can be eliminated by raising $\frac{\eta_1}{\eta_2}$ under ITS. A simple way to raise $\frac{\eta_1}{\eta_2}$ is **improving exploration**. Substituting the expression of $\eta_1$ and $\eta_2$ into Eq.7, we have

$$\epsilon > \frac{m}{(\frac{e_Q}{\alpha})^{\frac{1}{n-1}} + 1 + m - 1} \tag{8}$$

When $\boldsymbol{u}_s = argmax_{\boldsymbol{u}}\mathcal{Q}(s, \boldsymbol{u})$, we obtain the **lower bound of $\epsilon$** (denoted by $\epsilon_0$) from the right side of Eq.8. However, as the number of agents $n$ and the size of individual action spaces $m$ increases, $\epsilon_0$ grows close to 1 (as verified in Fig.1(b)), which is inapplicable in tasks with long episodes.

Another way to raise $\frac{\eta_1}{\eta_2}$ is **increasing the relative weights of the superior samples $\boldsymbol{u}_s$**. It is proved in Appendix E that the non-optimal stable points can be eliminated by applying a weight $w(w > \frac{\alpha(\eta_2 - \eta_1)}{e_Q \eta_1})$ to the superior samples under ITS. When $\boldsymbol{u}_s = argmax_{\boldsymbol{u}}\mathcal{Q}(s, \boldsymbol{u})$, we obtain the **lower bound of $w$** (denoted by $w_0$). However, the $w_0$ grows exponentially as the number of agents $n$ increases (as verified in Appendix E.2, $\boldsymbol{w_0 = 659.50}$ when $\boldsymbol{n = 4}$), which introduces instability in training.

Since $\frac{\eta_1}{\eta_2}$ decreases rapidly as the joint action space grows, the essence to raise $\frac{\eta_1}{\eta_2}$ is increasing the proportion of superior samples and **freeing the raised proportion from the dependency on the joint action space size**, for which we introduce the superior experience replay.

### 4.2 SUPERIOR EXPERIENCE REPLAY

To distinguish superior samples, we first introduce a critic $V(s)$, which approximates the true Q value of the joint greedy action. The target of the critic is defined as

$$\mathcal{V}_{gvr}(s) = \begin{cases} \mathcal{Q}(s, \boldsymbol{u}) & V(s) < \mathcal{Q}(s, \boldsymbol{u}) \ or \ \boldsymbol{u} = \boldsymbol{u}^* \\ V(s) & others \end{cases} \tag{9}$$

Inspired by prioritized experience replay (Schaul et al., 2015; Zhang & Sutton, 2017), we introduce a superior buffer where the episodic trajectories are stored. The number of superior samples which satisfy $\mathcal{Q}(s, \boldsymbol{u}) > V(s)$ and $\boldsymbol{u} \neq \boldsymbol{u}^*$ within a trajectory is defined as the priority of the trajectory. In the superior buffer, trajectories are ranked according to their priorities. The training batch consists of two parts: trajectories randomly sampled from the replay buffer and the top $k$ trajectories from the superior buffer. At the end of each episode, the trajectories sampled from the superior buffer will be put back after the update of their priorities. The working principle of GVR and the algorithm is given in Appendix G.

With SER, the superior samples are consist of two part: sampled from the superior buffer with a small probability and sampled from the superior buffer with a constant probability. As a result, the proportion of superior sample is mainly determined by the relative weight on samples from superior buffer, which is a constant and irrelevant to the joint action space size. It is proved in Appendix F that SER can eliminate the non-optimal stable points under ITS.

## 5 EXPERIMENTS

In this section, firstly we verify the attributes of stable points for linear value decomposition (LVD) in matrix games, where we also evaluate the improvements of GVR. Secondly, to evaluate the stability and scalability of our method, we test the performance of GVR in predator-prey tasks with extreme reward shaping. Thirdly, we conduct experiments on challenging tasks of StarCraft multi-agent challenge (SMAC) (Samvelyan et al., 2019). Finally, we design ablation studies to investigate the improvement of GVR. Our method is compared with state-of-the-art baselines including QMIX (Rashid et al., 2018), QPLEX (Wang et al., 2020), and WQMIX (Rashid et al., 2020). All results are evaluated over 5 seeds. Additional experiments are provided in Appendix I.

## 5.1 ONE-STEP MATRIX GAME

Matrix game is a simple fully cooperative multi-agent task, where the shared reward is defined by a payoff matrix. In one-step matrix games, the true $Q$ values are directly accessible from the payoff matrix, which is convenient for the verification of the optimal consistency.

**The verification of stable points for LVD.** We conduct experiments on two-agent one-step matrix game to verify the expression of joint Q values (i.e., Eq.3). We also verify the effect of reward function and exploration rate $\epsilon$ on the stable points. The experimental results and conclusions are provided in Appendix G.

**Evaluation of GVR.** Since the stability under ITS is determined by Eq.6, we first verify the expression under a random inferior true Q values ($\mathcal{Q}$). Two payoff matrices of size $3^4$ (i.e., m=3 and n=4) are generated for VDN and ITS respectively:

$$\mathcal{Q}(vdn) = \begin{cases} 6(1 + e_Q) & \boldsymbol{u} = \{0,0,0,0\} \\ 6 & \boldsymbol{u} = \{2,2,2,2\} \\ 6(1-\alpha) & others \end{cases} \quad \mathcal{Q}(its) = \begin{cases} 6(1 + e_Q) & \boldsymbol{u} = \{0,0,0,0\} \\ 6 & \boldsymbol{u} = \{2,2,2,2\} \\ random(-20,6) & others \end{cases} \quad (10)$$

where $e_Q = 0.3$, $\alpha = 0.1$. We measure $\Delta Q_{opt} = Q(0,0,0,0) - Q(2,2,2,2)$ for VDN and ITS trained with corresponding matrices, where the greedy action is fixed to $\boldsymbol{u}^* = \{2,2,2,2\}$. As shown in Fig.1(a), the tested $\Delta Q(\boldsymbol{u}_{opt}, \boldsymbol{\tau})$ **consists with our calculation result** ($\Delta Q(\boldsymbol{u}_{opt}, \boldsymbol{\tau}) = -1.951$) from Eq.6. Besides, $\Delta Q(\boldsymbol{u}_{opt}, \boldsymbol{\tau})$ of ITS with random inferior $\mathcal{Q}$ equals to that of VDN with fixed inferior $\mathcal{Q}$, which suggests that under ITS target, the stable points are **irrelevant** to the return of inferior samples. As a result, ITS largely frees the stable points from the interference of the reward function.

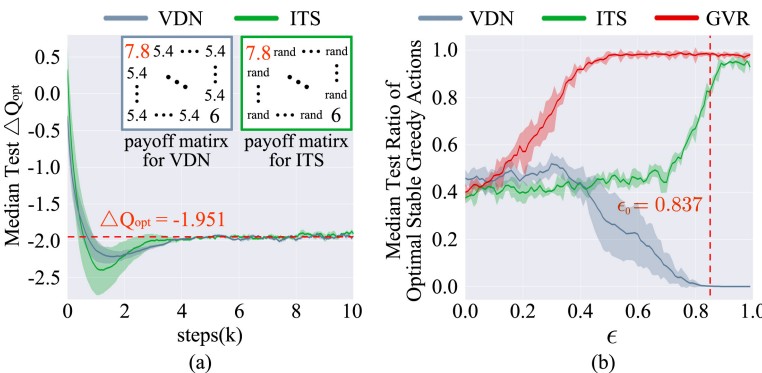

Figure 1: Evaluation of GVR in 4-agent matrix games. (a) Comparison of $\Delta Q_{opt}$ between ITS with random reward shaping and VDN with fixed reward shaping. The red dash line ($\Delta Q(\boldsymbol{u}_{opt}, \boldsymbol{\tau}) = -1.951$) denotes the calculation result (from Eq.6) under ITS target. (b) Median test ratio of the optimal stable points as $\epsilon$ grows (i.e., test probability of the optimal convergence), where $y = 1$ indicates the optimal stable point is the unique stable point and $y = 0$ indicates the optimal point is unstable. The red dash line ($\epsilon_0 = 0.837$) denotes the calculated lower bound of $\epsilon$ (from Eq.8) when ITS eliminates non-optimal stable points.

We also evaluate the ratio of optimal stable points for VDN, ITS and GVR as $\epsilon$ grows, where the payoff matrices are generated according to $Q(its)$ (Eq.10) over 5 seeds. At each value of $\epsilon$, 100 times of independent training and test are executed. According to Fig.1(b), for VDN, the optimal point become unstable (i.e., $y = 0$) when $\epsilon > 0.8$. For ITS, the **optimal points is always stable** (i.e., $y > 0$ ($\forall \epsilon \in (0,1]$)). Besides, ITS eliminates most of the non-optimal stable points under **large exploration** ($\epsilon > 0.837$), which **consists with our calculation result** (the red dash line) from Eq.8. For GVR, the optimal stable point is the unique stable point (i.e., the optimal consistency holds) under a wild range of $\epsilon$. However, GVR is unable to ensure the

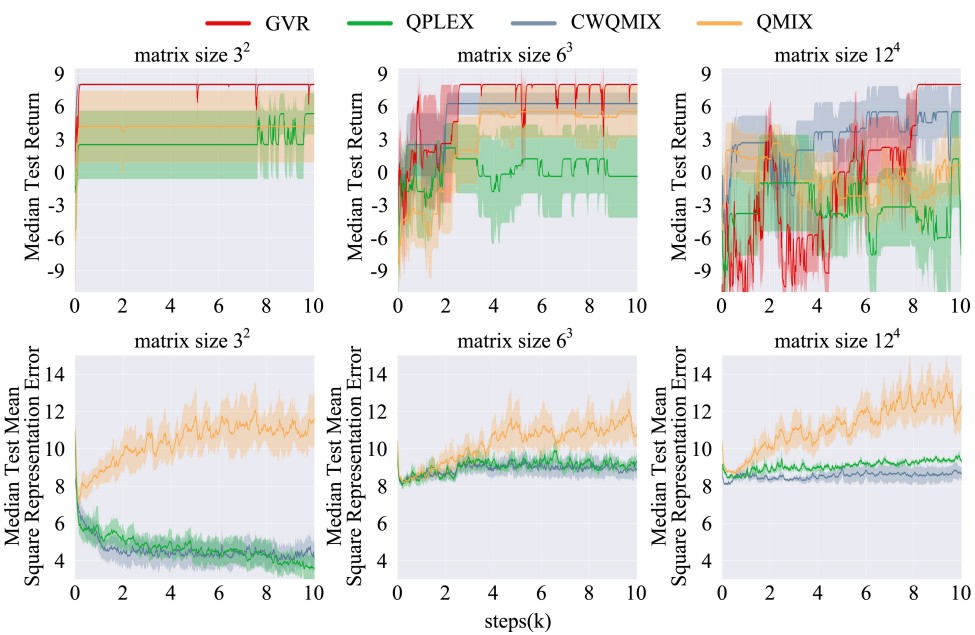

Figure 2: GVR vs methods with complete expressiveness capacity in matrix games.

optimal consistency under a small $\epsilon$, where the optimal sample is not explored in given training steps (e.g., the probability of the optimal sample $\boldsymbol{u}_{opt} = \{0, 0, 0, 0\}$ is $1.98e - 5$ under the greedy action $\boldsymbol{u}^* = \{2, 2, 2, 2\}$).

**GVR vs methods with complete expressiveness capacity.** To compare the efficiency between GVR and methods with complete expressiveness capacity, we evaluate GVR, QPLEX and CWQMIX in matrix games with different scales of action space. Similar to $Q(its)$ in Eq.10, we generate random matrices of size $3^2$, $6^3$, and $12^4$ over 5 seeds, where the first and the last element are set to be 8 and 6 respectively. From Fig.2, in the last two tasks ($U^n = 6^3$ and $U^n = 12^4$), the representation errors of CWQMIX and QPLEX do not decrease during training, which suggests that they are unable to learn the complete expressiveness within given steps. We do not measure the representation error of GVR because the representation target is modified by ITS. GVR is the only method ensuring the optimal coordination (i.e. median test return $= 8$).

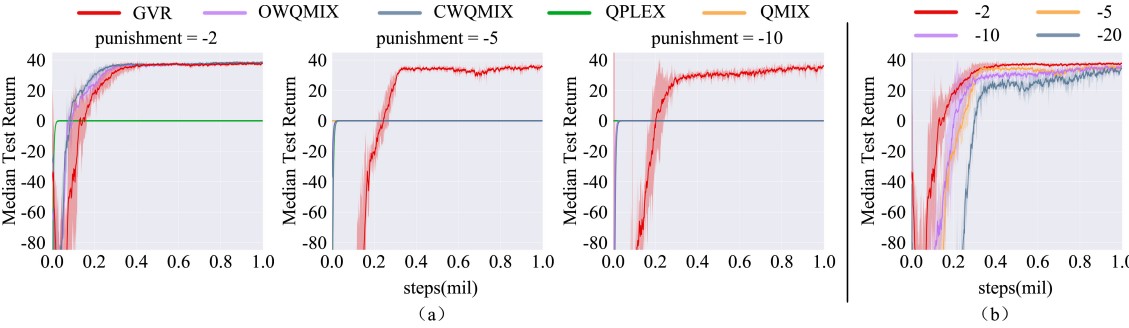

Figure 3: Experiment results on predator-prey. (a) Comparison between GVR and baselines. (b) Performance of GVR with different punishments.

## 5.2 PREDATOR-PREY

In this subsection, we compare GVR with state-of-the-art baselines in the predator-prey tasks (Böhmer et al., 2020). In our experiments, 8 predators are trained to capture 8 preys, where the preys are controlled by random policies. The reward is shared by all predators, and a punishment is applied to the reward when only a single agent capture a prey. Our experiments are carried out under 3 punishment values.

From Fig.3, VDN and QMIX fail in all experiments, where all agents tend to avoid the preys. QPLEX also fails in spite of its complete expressiveness capacity. WQMIX can only solve the task with a small punishment of -2, while GVR is able to solve the tasks under all punishments.

## 5.3 STARCRAFT MULTI-AGENT CHALLENGE

StarCraft multi-agent challenge (SMAC) is a popular benchmark for the evaluation of MARL algorithms. We compare GVR with baselines in various difficult tasks of SMAC. The game version is 69232. Each algorithm is trained for 2e6 steps in MMM2, 2c_vs_64_zg and 6h_vs_8z, with $\epsilon$ damping from 1 to 0.05 during the first 5e4 steps. Especially, in 6h_vs_8z, each algorithm is trained for 5e6 steps, with $\epsilon$ damping from 1 to 0.05 during the first 1e6 steps.

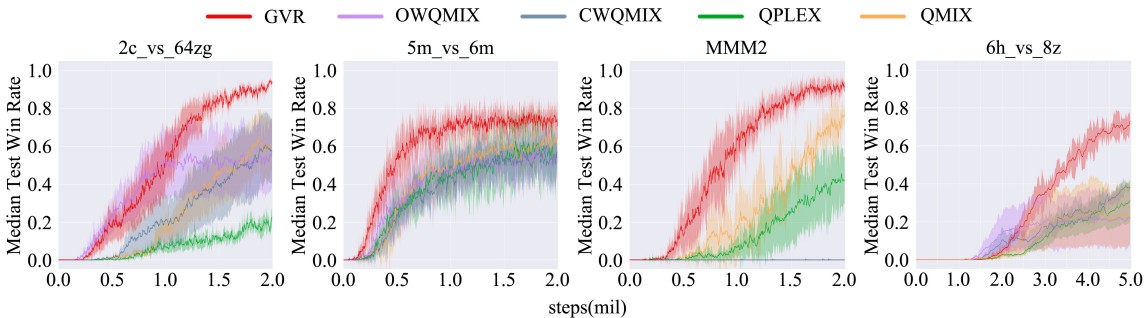

Figure 4: Median test win rate vs training steps.

From Fig.4, GVR shows the best performance. Different from predator-prey with abnormal punishments, the reward function in SMAC is more reasonable, where the linear and monotonic value decomposition can meet the TGM condition approximately. As a result, the algorithms with full representation expressiveness capacity (QPLEX, OWQMIX, CWQMIX) do not perform better than QMIX due to the difficulty of complete expressiveness.

## 6 CONCLUSION AND FUTURE WORK

This paper discusses the optimal coordination in fully cooperative MARL tasks and proposes a new criterion (i.e., optimal consistency) to evaluate the optimality of coordination in value decomposition. To achieve the optimal consistency, we introduce the TGM condition for linear and monotonic value decomposition, where it is proved the TGM can be ensured if the optimal stable point is the unique stable point. Therefore, we propose the GVR algorithm. Which ensures the optimal stable points via ITS and eliminates the non-optimal stable points via SER. In experiments on matrix games, we verify our calculation results, which directly demonstrates the effect of GVR. Besides, in experiments on predator-prey and SMAC, GVR outperforms state-of-the-art baselines. However, GVR is unable to ensure the TGM condition in hard exploration tasks, where the superior samples are difficult to obtain. We are interested in the combination of GVR with efficient exploration approaches in future work.

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

•

