# OpenReview forum: "Greedy-based Value Representation for Efficient Coordination in Multi-agent Reinforcement Learning"
_ICLR.cc/2022/Conference — ICLR 2022 Submitted_

### Official Review · Reviewer_bdoB · 2021-11-02

**Correctness:** 2
**Technical Novelty And Significance:** 3
**Empirical Novelty And Significance:** 2
**Recommendation:** 3
**Confidence:** 4

**Main Review:**

### Strengths:

* The ideas in the paper are novel, and experimental results validate the equations as well. Moreover, the ITS method is based on a proof and can guarantee the stability of greedy actions.
* The experimental results on some super hard SMAC maps like 6h_vs_8z (known to be a hard exploration map) look impressive compared to other state of the art methods like QMIX, QPLEX.
* The paper is well motivated as both IGM and TGM conditions are essential for stability and learning coordinated policies, and can be especially helpful for non-monotonic tasks.

### Weaknesses:

* ITS can keep the greedy action stable which works great if the greedy action is already optimal. However, the authors rely on superior experience replay to destabilize the non-optimal greedy actions. I have some major concerns about this and would request authors to clarify the following:

	- Looking at Algorithm 1 in appendix, it is unclear how the agent loss is computed. If the Q value of an action $u$ is greater than the greedy action $u^\star$ i.e. $Q(s,u) > Q(s,u^\star)$, then it seems like the error would be zero as $Q^{its}(s,u) = Q(s,u)$. Does this rely on the greedy action always being the optimal joint action through higher exploration rate or superior experience replay?

	- Similarly, the update for the critic is a bit unclear as well. Why is the critic loss only updated for superior samples? It makes sense to consider the samples with $Q(s,u) > V(s)$ or $u = u^\star$ as superior samples, but in the beginning of the training, the $Q(s,u) < V(s)$ can happen for samples with true optimal joint actions as well, but the critic loss will not be updated for them.

* The paper is a bit hard to follow in certain areas, especially Section 5.1. It would be great if the authors could take some time to add more details to the paper as follows.

	- In Section 5.1, the authors can make it easier to understand what VDN+ITS variant refers to.
	- The paper sometimes refer to $u^\star$ as optimal greedy  joint action (in Section 4.1), and then refer to $u^\star$ as just greedy joint action in section 5.1 which is a bit confusing.
	- Should Equation (3) have $Q_{ik}$ and $Q_{kj}$ instead of $Q_{ki}$ and $Q_{jk}$?

* One weakness of this paper is that the experimental section is missing  some important baselines. The destabilization of the non-optimal greedy actions through the superior replay buffer is interesting, but the authors should empirically compare against other efficient exploration based MARL approaches such as [1,2,3,4]. Although these approaches also cannot guarantee both IGM and TGM, they can potentially also alleviate the destability associated with stable non-optimal greedy actions via different exploration mechanisms.

* I have a major concern about the performance of WQMIX reported on the StarCraft II benchmark map 6h_vs_8z. The reported performance is not consistent with that reported in the WQMIX paper (see Figure 4 in [5]). WQMIX paper also uses the same exploration schedule as this paper i.e. $\epsilon$ annealed over 1e6 time steps. The original paper reports a median test win rate of around 80% with optimistic weighting but this paper only reports around 20%. Also, there seems to be only a single seed used for evaluating GVR on 6h_vs_8z.

* The authors seem to have skipped a few super hard SMAC maps such as corridor, 27m_vs_30m and 3s5z_vs_3s6z which are important to determine the scalability of their approach with more agents.

* The code is missing from the submission for reproducibility.

### References:

[1] Wang, Tonghan, et al. "Influence-based multi-agent exploration." arXiv preprint arXiv:1910.05512 (2019).

[2] Liu, Iou-Jen, et al. "Cooperative exploration for multi-agent deep reinforcement learning." International Conference on Machine Learning. PMLR, 2021.

[3] Mahajan, Anuj, et al. "Maven: Multi-agent variational exploration." arXiv preprint arXiv:1910.07483 (2019).

[4] Gupta, Tarun, et al. "Uneven: Universal value exploration for multi-agent reinforcement learning." International Conference on Machine Learning. PMLR, 2021.

[5] Rashid, Tabish, et al. "Weighted qmix: Expanding monotonic value function factorisation." arXiv e-prints (2020)

**Summary Of The Paper:**

This paper aims to improve value-factorization in cooperative multi-agent reinforcement learning settings under centralized training and decentralized execution (CTDE) framework. The proposed method (GVR) attempts to ensure both Individual-Global-Max(IGM) and True-Global-Max(TGM) conditions without learning a completely expressive (CEC) value function (which can represent all joint- actions). IGM ensures the consistency between joint action selection and local greedy action selection (Individual-Global-Max, IGM), and TGM ensures that the joint-action value function correctly represents optimal values. The GVR method proposed by the authors claims to ensure both IGM and TGM conditions without CEC.

The main contribution is greedy-value based representation (GVR) which consists of two parts. First, Inferior Target Sampling (ITS) ensures that the greedy joint action is stable such that if the greedy action corresponds to the optimal joint action, then for other joint actions, the gradient will be negative, thereby selectively focusing on representing the unique optimal joint action. If the greedy action does not correspond to the unique optimal joint action, then higher exploration is required in order to destabilize it. However, the lower bound for exploration can become quite high (close to 1) with increasing agents/joint action space, therefore the authors propose a prioritized experience replay buffer. The prioritized buffer assigns higher priority to non-greedy actions which have higher values than a state-based critic.

Experimental results on matrix game, predator-prey and hard/super-hard maps from Starcraft II benchmark show improvements over baseline methods.


**Summary Of The Review:**

The paper is well motivated so as to ensure both IGM and TGM conditions, and therefore stable convergence in MARL. The ideas in the paper are novel, and ITS is based on a proof and can guarantee the stability of greedy actions.  However, I am not convinced that the proposed superior experience replay can destabilize the non-optimal greedy actions.  I have asked a clarification question about this and will improve my score based on a satisfactory response. Other than this, although the empirical results are good, the authors have missing baselines from the experimental evaluations, and missing experiments with respect to proving the scalability of GVR with more agents. Finally, I have concerns regarding evaluations reported on 6h_vs_8z. Therefore, I don’t think the paper is ready for publication as is.

---

> ### Author Response · Authors · 2021-11-10
> **Replies**
>
> Thanks for your questions and comments, we list the replies as follows
>
> 1.	Target of the agents and critic
> We have further clarified our method in the introduction and Section 4.1. We also provide more theoretical and empirical proves of ITS and SER in Appendix D1, E and F.
>
> (1)	The ITS target for $\mathcal{Q}(s,u)> Q(u^*,\tau)$ is $\mathcal{Q}(s,u)$. The error of joint Q value function is $\mathcal{Q}(s,u)-Q_{\theta_a}(s,u)$, which is not zero.
>
> (2)	Notice one of the conditions to update the critic is $u=u^*$, i.e., the critic is updated when receiving a greedy action. If $\mathcal{Q}(s,u)<V(s)$ in the beginning, $V(s)$ will approximate the true Q value of current greedy action (i.e., $\mathcal{Q}(s,u^*)$). The critic is designed to distinguish whether an action is better than the current greedy action. Therefore, we let the critic approximate the true Q value of the greedy action. In practice, the critic is also updated when receiving a superior sample. As a result, the output of the critic is marginal above the true Q value of the greedy action, and the superior sample with a slight advantage will be ignored. This is helpful for the stability of training.
>
> 2.	details
>
> (1)	VDN+ITS refers to the method where we simply apply the ITS target to VDN, which is modified to ITS in the resubmitted version. The comparison between VDN and VDN+ITS is used to directly verify the effectiveness of ITS target.
> (2)	We have clarified the notations in resubmitted preliminary.
> (3)	Thanks for pointing out this typo, we have revised it in resubmitted version.
>
> 3.	baselines
>
> We build connections between efficient exploration based approaches and our method, detailed information can be found in Appendix A. The comparison between these methods and our method is provided in Appendix I.2. However, two baselines (CMAE and EDTI) are not included in our experiments because both methods are count-based, which is impractical to apply them to complex tasks in SMAC.
>
>
> 4.	the results of WQMIX
>
> WQMIX seems to perform unsteadily under different seeds (e.g., in our experiments on 2c_vs_64zg, the test win rate of WQMIX varies from 0 to 0.8). The authors of WQMIX do not provide the seeds of their curves. Despite we have downloaded the source code from their website, we can not reproduce their experiment results.
>
>
> 5.	Super hard experiments on SCII
> We have complemented the experiments on corridor and 3s5z_vs_3s6z in Apendix I.2. Due to the memory limitation of our machine, we can not implement the experiments on 27m_vs_30m.
>
>
> 6.	code
> Key resources and details (proofs, experimental setup) are sufficiently described to reproduce the main results (especially the results on matrix games). And we will post the code on github as soon as possible.
>
>
> Besides, we notice that the reviewer mentioned in the summary that he/she is not convinced that our method can destabilize the non-optimal greedy action. Therefore, we provide proof in Appendix F in resubmitted version.

---

### Official Review · Reviewer_Nsqb · 2021-11-02

**Correctness:** 2
**Technical Novelty And Significance:** 2
**Empirical Novelty And Significance:** 2
**Recommendation:** 3
**Confidence:** 3

**Main Review:**

Strength
    The paper introduces new condition TGM for value deposition MARL,and the reviewer thinks that this condition should continue to be discussed in the value deposition method. The experiment for ablation study shows their claim well, and the performance of the proposed method GVR is shown to be outperformed than the state-of-the-art value decomposition methods in various experimental environments.

    Weaknesses
    1. In the paper, the authors used true $Q$ function and joint $Q$ value function, but reviewer doesn't find the definition of joint $Q$ value function. What is the difference between joint $Q$ value function and true $Q$ function?.

    2. In the equation (3), the authors expressed the utility function as true $Q$ function. If joint $Q$-value function is used for equation (3) instead of true $Q$ function, then the equation is true. However, the author use true $Q$ function and reviewer doesn't understand how to derive the equation (3). Some more explanations are needed

    3. In Appendix D and E, the authors prove their claim, but I couldn't follow the process of proof. How can the author express the utility function as equation (13) in appendix D? The reviewer doesn't understand how mapping $f$ occurs. Therefore, more explanation is needed to understand equations 13 and 18

    4. Finally, the authors madeThe author proposes a new additional condition for value decomposition in MARL, True-Global-Max (TGM) condition, which is reasonable in some respects, but the reviewer believe that in the proof of the author's claim, there are lots of explanation to understand. Thus, if the author can solve the above mentioned questions, the reviewer will raise the score. target of critic $V(s)$ as equation (9), but there is no explanation of why that can happen. You need explanation of reason for target of critic.

**Summary Of The Paper:**

The authors introduced the optimal consistency for value decomposition methods in multi-agent reinforcement learning (MARL) and suggests True-Global-Max (TGM) condition along with Individual-Global-Max (IGM) proposed by QTRAN to achieve the optimal consistency. Then the authors suggested the greedy-based value representation (GVR) through the inferior target shaping and superior experience replay to ensure the TGM condition.


**Summary Of The Review:**

The authors proposed a new additional condition for value decomposition in MARL, True-Global-Max (TGM) condition, which is reasonable in some respects, but the reviewer believe that in the proof of the author's claim, there are lots of explanation to understand. Thus, if the authors can solve the above mentioned questions, the reviewer will raise the score.

---

> ### Author Response · Authors · 2021-11-10
> **Replies**
>
> Thanks for your questions and comments. We list the replies as follows
>
> To clarify our methods and derivations, we made great modifications in the resubmitted paper. Therefore, we denote the equations and appendixes of resubmitted version with $(R)$ (e.g., Appendix B(R)) in replies.
>
> 1. The joint Q value function is the action-value function of the agent group, which is modeled by neural networks. The true Q value is the actual action-value of the agent group, which is the external criterion of the team's performance defined by the environment. The target of the joint Q value function is just the true Q value.
>
> 2. For the question about Eq.3 (i.e., Eq.2 of Appendix B(R)), the utility function equals to the expectation on its target (e.g., $\mathcal{Q}_{ik} - \mathcal{U}_k^2$ is a target of $\mathcal{U}_i^1$ with a probability $\epsilon/m$). If we replace true Q values with the joint Q values,  according to Eq.1 of Appendix B(R), the equation still holds. In fact, the equation holds for both situations.
>
> 3. Eq.13 in Appendix D (i.e., Eq.19 of Appendix D.2(R)) is acquired with the following steps: (1) refer to the expression of the utility function in Eq.2 (Appendix B.1(R)); (2) replace all true Q values of actions except the greedy action and $u_s$ with the target $(1-\alpha)Q(u^*, \tau)$; (3) combine the terms containing utility functions and use the mapping $f$ to represent it. The accurate expression of the mapping $f$ is very complicated, which is unnecessary because it would be canceled out by a subtraction (Eq.23, Appendix D.2(R)). We also provide proof without the mapping $f$ in Appendix D.1(R).
>
> For Eq.18 in appendix E, we update the proof to n-agent situation. The high-level idea is to represent the utility function with the expectation on different kinds of samples (i.e., inferior, current and greedy). Please refer to Appendix E.1(R) for the details.
>
> 4. The critic is designed to distinguish whether an action is better than the current greedy action. Therefore, we let the critic approximate the true Q value of the greedy action. In practice, the critic is also updated when receiving a superior sample. As a result, the output of the critic is marginal above the true Q value of the greedy action, and the superior sample with a slight advantage will be ignored. This is helpful for the stability of training.

---

> > ### Comment · Reviewer_Nsqb · 2021-11-30
> > **Response**
> >
> > Thank you for the author's responses.
> >
> >  (1) In author's response, author says the joint Q function is modeled by neural network. Then I think the utility funcitons in Eq(3) should also be modeled by neural network so we don't know the utility function. However, the author makes the utility function as expactation of $Q_{ik}-U_k^2$. I can't understand how you can ensure this expression. Furthermore, the proposed target of utility function is not reasonable, and I think the target of true utility function may be considered as expectation of true Q function over other agent's action i.e $\frac{\epsilon}{m}\sum_{k=1}^m Q_{ik} + (1-\epsilon)Q_{ij^{\ast}}$
> >
> >  (2) The proof in Appendix D is based on the fact the utility function equals to the expectation on the proposed target of utility function, but as metioned in (1), I can't agree this.
> >
> >  (3) I can understand what the author is trying to say in target of critic, but the proof is not acceptable to me.
> >
> >  (4) In the proposed algorithm, I can't find the usage of the reward function. In the both critic loss and agent loss, the algorithm doesn't use the reward in the target becuase the author makes target as equation (5) and (9). I didn't understand why the proposed algorithm can work without reward.
> >
> > To due the above reasons, the paper is still not enough to raise the score.

---

### Official Review · Reviewer_dbgG · 2021-11-02

**Correctness:** 2
**Technical Novelty And Significance:** 4
**Empirical Novelty And Significance:** 4
**Recommendation:** 3
**Confidence:** 3

**Main Review:**

The authors seem to attempt a deep theoretical analysis of the underlying problem and the good results on predator-prey and StarCraft II indicate that they have found an algorithm that should be definitely published. However, although the reviewer has published on this field, he/she was largely unable to follow the text and the derivations. While the text is technically well written, the underlying concepts are very unclear to the reviewer and the formulas are swapping notations. Finally the developed algorithm remained unclear to the reviewer, who is not sure how statements like (eq.6) can actually be implemented (the pseudo-code in the appendix did not help). In detail:

(1) As far as the reviewer understands the field, the problem is that monotonic value function functions cannot represent non-monotonic true values. Rashid et al. (2020) have shown that putting more importance on optimal actions can resolve this issue and Gupta et al. (2021) have linked this to relative overgeneralization, showing that sampling optimal actions more often allows to learn the optimal policy. Both papers use matrix-games similar to Table 1 to motivate their approach. However, the reviewer was not able to connect these insights with the notion of a "stable greedy action". Which are those? Definition 3 uses the formulation "a stable joint Q value function $Q(s,u)$ which has converged to the true Q value $\mathcal Q^\pi(s, u)$ under $\pi(s, u)$". Does this mean that $Q(s,u) = \mathcal Q^\pi(s, u)$? Is $Q$ factored? How are stability and factoredness related? This seems to be the central concept of GVR, but the reviewer has no idea how to connect it to the presented problem or the proposed solution.

(2) The core of the ITS method seems to be (eq.6). How is it computed in practice, when the agent does not know $\mathcal Q(s, u)$, which the reviewer assumes corresponds to $\mathcal Q^\pi(s, u)$? For the current action this could be approximated with the Bellman operator, but (eq.6) requires  the evaluation of $\mathcal Q(s, u^*)$ as well. It would be generally helpful to differntiate between $u^*$ and $u_{opt}$ throughout the text, but in particularly here, where the agent does not know $u_{opt}$, but uses $u^*$ in (eq.6), which is confusingly defined as "optimal".

(3) The formal derivation is at least sloppy. The authors introduce the Dec-POMDP framework where the policy $\pi(u|\tau)$ conditions on the action-observation histories, but then define the utilities $\mathcal U^a(u^a, o^a)$ as conditioning only on the observations $o^a$, and the values $Q(u, s)$ as conditioning on the state. This does not work (see Oliehoek and Amato, 2016). Either work in a Dec-MDP, which ignores many interesting issues, or consistently condition on the histories $\tau^a$ and not the state or observation.

(4) The paper uses the term "reward shaping", but it seems the authors mean "reward function" (as in the matrix game experiments). This is very confusing, as GVR "shapes" the value-targets, which one could interpret as actual reward shaping, i.e., (here indirectly) changing the given reward function to improve performance.

(5) (eq.8) insinuates that higher exploration noise will "destabilize" non-optimal stable greedy policies. How does this fit with the relative overgeneralization example in Gupta et al. (2021)?

(6) The reviewer did not understand the conclusions the authors drew from the matrix game experiment. Both the setup (why are the rewards for most actions randomized? how is this a fair comparison?) and the results in Figure 2 (are there siginificant differences in return?). While Figure 1b seems to support the theoretical statement, it is unclear why the setup justifies this (e.g. what would happen if your random rewards are from another range than (-20,6)?).

**Summary Of The Paper:**

The paper addresses the problem of monotonic value representations for non-monotonic true values in MARL. The authors perform a theoretical analysis of the conditions under which the greedy decentralized policy coincides with the optimal joint policy and derive a novel update scheme to destabilize incorrect fix-points. They also introduce a priority replay buffer that selects transitions with optimal actions more often and thereby stabilizing learning further. The new algorithm GVR is tested on a matrix game, a predator-prey task and StarCraft II micromanagement tasks. In the latter two GVR appears to significantly outperform decentralized baselines.


**Summary Of The Review:**

The paper seems to be onto something that would be of great interest to the community, and the good results indicate that the authors know what they are doing, but the paper is currently almost incomprehensible to this reviewer. Although the authors are welcome to refute or explain the above criticisms, the fact that the reviewer was not able to understand the core idea of the paper during the first reading makes it unlikely that he/she will recommend to publish it in the current form. The reviewer would like to encourage to the authors to rewrite and resubmit the paper, though, as the content seems to be truly significant!


**POST-REBUTTAL**

Thanks to the authors for their clarifications. However, it was not enough to clarify the paper's main message to the reviewer. The reviewer will therefore not change the score.

---

> ### Author Response · Authors · 2021-11-10
> **Replies**
>
> Your comments and questions are really helpful to us, and we have thoroughly revised our paper. We list the replies for the questions as follows
>
> 1. According to the investigation on LVD and MVD (Section 3.2 of resubmitted paper), the non-optimal stable point is the root cause of non-optimal coordination and relative overgeneralization.
>
> We connect our conclusion with recent works in the resubmitted version. Both methods the reviewer mentioned (Placing more importance on the optimal action or sampling optimal action more often) increase the proportion of the optimal sample, which helps to eliminate the non-optimal stable points (as proved by Eq.7). However, these methods can not ensure convergence to the optimal coordination (i.e., optimal consistency). More details are provided in Appendix A.
>
>
> Due to the representation limitation of LVD and MVD, the joint Q value function can not fully represent the true Q values. As a result, the converged joint Q values are not equal to the true Q value. Instead, they converge to the situations in Tab1(b) or Tab1(d). $Q$ is linearly (or monotonically) factorized. However, such factorization introduces representation limitation, which causes multiple stable points. We clarify Def.3 and present an example to explain the stability in Section 3.2 of resubmitted paper.
>
>
> 2. Firstly, the reviewer is right about the assuming of $\mathcal{Q}(s,u)$. For Eq.6 (Eq.5 in resubmitted version), we mistyped $Q(u^*,\tau)$ with $\mathcal{Q}(s,u^*)$, which has been revised in the resubmitted version. We denote the greed action with $u^*$ and optimal action with $u_{opt}$ throughout the paper, which is defined in the resubmitted preliminary.
>
>
> 3-4.	 Thanks for your kind and insightful comments, we have revised these problems. In addition, to clarify our methods and derivations, we have modified the notations in the resubmitted version.
>
>
> 5. In UneVEn, the authors analyze a special case (a well-shaped two-agent payoff matrix game) to support the idea that the convergence to the optimal joint action depends on the probability of the other agent's optimal action. The conclusion is consistent with our findings (Eq.7), where we consider a more general situation (n-agents, any reward function).
>
>
> 6. We generate the matrices randomly for two reasons. First, our experiments on matrix games are carried out in a larger action space (up to $12^4$) than previous work, where the manual design of the matrix is time-consuming. Second, the non-optimal stable points depend on reward function (as verified in Appendix H), to evaluate whether a method can eliminate the non-optimal stable points (or ensure optimal consistency), we test this method under general (random) reward functions. To guarantee the fairness of comparison, We use the same set of seeds for all methods to generate the matrices (i.e., the generated matrices are the same for all methods).  The difference in return in figure 2 may be not significant, but we mainly concern the stability and optimality of evaluated methods. In figure 2, at the end of the training, our method converges to the optimal (return = 8) with little variance. The range of the random reward is not necessarily (-20,6), it can be any other range. As verified in the experiment on predator-prey, the reward function has little influence on our method.

---

> > ### Comment · Reviewer_dbgG · 2021-11-29
> > **Still not clear enough**
> >
> > Thanks to the authors for their clarifications. However, it was not enough to clarify the paper's main message to the reviewer. The reviewer will therefore not change the score.
> >
> > The reviewer admits that he/she did not spend the same time reading the revised script, but the story-line is still very confusing. In particular the formal side is still unclear. For example, what does "with target $\mathcal Q$" (which should throughout the text be $\mathcal Q^\pi$) mean? How is the target approximated? Is that important at all?
> >
> > Furthermore, there are notational inconsistencies: the true value $\mathcal Q^\pi(s,a)$ must also condition on all histories. As the histories can not be deduced by the state, different histories would mean different futures and thus different values.
> >
> > The reviewer recommends that the authors take a step back and completely rewrite the argumentation in the paper. This is not just a question of mathematical correctness, but also of intuitive understanding for someone who is familiar with prior work, where the key to learning the optimal policy seemed to be the update distribution.

---

### Official Review · Reviewer_LsGH · 2021-11-04

**Correctness:** 4
**Technical Novelty And Significance:** 3
**Empirical Novelty And Significance:** 3
**Recommendation:** 6
**Confidence:** 2

**Main Review:**

Strengths:
The problem that is being studied is of high importance, given the popularity of the value decomposition baselines (VDN and QMIX). The True-Global-Max condition is interesting, and the paper proposes relatively simple techniques to satisfy the condition, which is a plus. The experiments are also interesting and the ablation studies in the supplemental are informative. I found the derivations and the writing easy to read, and overall I think the paper can be a good contribution.

Some questions I have:
- Def 3 and Tab1: the condition says that the joint Q function has converged to the true Q function under policy \pi. But in Table 1 it seems that none of the Q functions have converged to the true Q function (Tab 1a)? Could you please clarify Def 3 (I'm not sure why we can't just write Eq5 in terms of argmax Q^\pi = argmax \pi
- Is the joint interactive policy \pi fixed? Or is it updating as the estimated Q function updates? I assume it is updating, but that also seems to mean Eq5 will always hold?
- For inferior target shaping (ITS), we are penalizing non-optimal actions. The high-level idea is to have a large enough epsilon\*penalty so that the agents can realize that there is a better alternative action, right? But it is unclear to me why we don't just need that epsilon\*penalty > 0, it seems that if there is just a little improvement then over time the value iterations will converge to the better alternative?

**Summary Of The Paper:**

The paper studies the problem of value decomposition, which decomposes the joint-Q function into some linear or monotonic transformations of individual factored Q functions for each of the agents. The paper identifies limitations with previous linear/monotonic forms of the decomposition, in particular that the joint greedy action (which matches the greedy joint action for these decompositions) might not match the maximum true Q value. This condition is termed "True-Global-Max", and the paper introduces two techniques (inferior target shaping, superior experience replay) to satisfy this condition and improve upon previous suggested Q-value decompositions in literature. The paper walks through a toy matrix game example, a predator-prey experiment, and the Starcraft Multi-Agent challenge environment.

**Summary Of The Review:**

Overall I found the problem important, and the paper interesting (both technical and experimental parts) and easy to read. I do have some questions about the framework, and I would be more certain of my recommendation if the authors can help me clarify my confusion.

---

> ### Author Response · Authors · 2021-11-23
> **Replies**
>
> Thanks for your questions and comments, the answers are listed as follows
>
> 1. Due to the representation limitation of LVD and MVD, the joint Q value function can not fully represent the true Q values. As a result, the converged joint Q values are not equal to the true Q value. Instead, they converge to the situations in Tab1(b) or Tab1(d). We clarify Def.3 and present an example to explain the stability in Section 3.2 of resubmitted paper.
>
> 2. Whether the joint interactive policy is fixed or not depends on the situation. When we analyze or evaluate the stability, we fix the joint interactive policy (e.g., Tab.1, Fig.1(a)). When we evaluate the performance, the joint interactive policy is not fixed (e.g., Fig.1(b), Fig.2-Fig.4). For the question about Eq.5 (Eq.4 in the resubmitted paper), please refer to reply 1.
>
> 3. Indeed, the high-level idea is to make agents realize that there is a better alternative action. But I don't understand the "penalty", would you mean $\alpha$ (in Eq.5 of the resubmitted paper)?  A small value of $\alpha$ may lead to confusion between greedy and inferior actions.  Besides, we have proved in the resubmitted paper that a small improvement is insufficient to achieve the optimal consistency, there is always a lower bound for parameters (e.g., $\epsilon$ in Eq.8 of Section 4.1 and sample weight in Eq.33 of Appendix E.1).

---

### Decision · Program_Chairs · 2022-01-20

**Decision:**

Reject

**Comment:**

The paper studies the join-Q value decomposition problem in MARL. Some of the results are interesting, e.g., the True-Global-Max condition and several experiments. However, the majority of the reviews are negative due to the current presentation of the paper. We encourage the authors address all the reviewers' comments and submit a new version to the next conference.